# Impact of Preparticipation Hypohydration on Cognitive Performance and Concussion-like Symptoms in Recreational Athletes

**DOI:** 10.3390/nu15204420

**Published:** 2023-10-18

**Authors:** Anna Strüven, Stefan Brunner, Georges Weis, Yannick Cohrdes, Stephan Lackermair, Jenny Schlichtiger, Antonia Kellnar, Korbinian Lackermair

**Affiliations:** 1Department of Medicine I, University Hospital Munich, Ludwig Maximilian University, 81377 Munich, Germany; 2Center for Sports Medicine, University Hospital Munich, Ludwig Maximilian University, 81377 Munich, Germany; 3Department of Neurosurgery, Krankenhaus Barmherzige Brüder Regensburg, 93049 Regensburg, Germany

**Keywords:** exercise, dehydration, recreational athletes, concussion, SCAT3

## Abstract

Background: Sports-related concussion is a relevant risk of contact sports, with several million cases per year worldwide. Prompt identification is crucial to prevent complications and late effects but may be impeded by an overlap with dehydration-associated impairment of cognitive function. Researchers have extensively studied the effects of pronounced dehydration in endurance sports, especially in the heat. However, little is known about the effects of isolated and mild dehydration. Methods: Healthy recreational athletes underwent a standardized fluid deprivation test. Hypohydration was assessed by bioelectrical impedance analysis (BIA) and laboratory testing of electrolytes and retention parameters. Participants underwent cardiopulmonary exercise testing (CPET) with a cycle ramp protocol. Each participant served as their own control undergoing CPET in a hypohydrated [HYH] and a euhydrated [EUH] state. Effects were assessed using a shortened version of Sport Concussion Assessment Tool 3 (SCAT3). Results: Fluid deprivation caused a mild (2%) reduction in body water, resulting in a calculated body mass loss of 0.8% without alterations of electrolytes, serum-osmolality, or hematocrit. Athletes reported significantly more (1.8 ± 2.2 vs. 0.4 ± 0.7; *p* < 0.01) and more severe (4.4 ± 6.2 vs. 1.0 ± 1.9; *p* < 0.01) concussion-like symptoms in a hypohydrated state. Balance was worse in HYH by trend with a significant difference for tandem stance (1.1 ± 1.3 vs. 0.6 ± 1.1; *p* = 0.02). No relevant differences were presented for items of memory and concentration. Conclusions: Mild dehydration caused relevant alterations of concussion-like symptoms and balance in healthy recreational athletes in the absence of endurance exercise or heat. Further research is needed to clarify the real-life relevance of these findings and to strengthen the differential diagnosis of concussion.

## 1. Background

Physical and cognitive performance in athletes depends on numerous internal and external factors. An essential negative influence is dehydration [1]. Dehydration in athletes results from long-lasting training sessions or competition, primarily when carried out in the heat. 

The majority of the most popular worldwide sporting events (e.g., the FIFA World Cup, Summer Olympics, Tour de France, New York Marathon) are held during summer months. Exercise in the heat increases physical effort for different reasons: First, heat increases the thermoregulatory burden [2,3,4,5]. To avoid a rise in body core temperature, skin perfusion is elevated resulting in a reduced cardiac output accessible for musculature. Additionally, increased skin perfusion might reduce cardiac preload and thereby cardiac output [2,3,4,5,6,7]. Second, a heat-induced rise in sweat rate can lead to changes in plasma osmolality and dehydration if fluid and electrolyte loss is not compensated. Reduction in circulating plasma volume might impede cardiac output and changes in osmolality and, as a result, an imbalance of electrolytes and increased core temperature might deteriorate metabolic processes that form the basis of exercise capacity [4,5,8,9,10]. The physiological adaptation process during exercise in a high temperature environment is complex, which makes it difficult to identify and evaluate the differences in the impact of potential confounders on exercise capacity [5]. 

It has been shown that heat, especially without acclimatization, might impair cardiopulmonary exercise capacity [2,3,4,5,11,12,13,14,15,16,17]. Nevertheless, most studies evaluated the effects of dehydration on exercise capacity in models of exercise-induced dehydration (with dehydration resulting from intense sweating) or dehydration after heat exposure. These models provide several physiological alterations and impede the attribution of effects to fluid deficiency. 

To study the impairment of exercise capacity in dehydration mechanistically, it is important to use a model of “isolated dehydration”, i.e., dehydration without fatigue after long-lasting exercise, intense sweating (with concomitant changes in electrolytes, metabolism, etc.) or heat exposure as dehydration is intertwined with these in real life conditions. 

Previously, we could demonstrate in 50 recreational athletes that mild isolated dehydration (=hypohydration), resulting from pre-participation fluid deprivation, impaired aerobic exercise capacity, which was primarily explainable by a reduction in cardiac output [5]. 

Many sports like American football, soccer, tennis, or basketball feature rapidly changing intensity and require (aerobic) exercise capacity combined with repeated high-intensity bursts alternating with low intensity situations. Besides endurance, these sports also require reliable cognitive performance like coordination, concentration, or balance for the execution of complex skills or for decision-making [1]. 

Besides impairments of exercise capacity, dehydration is also known to impair these cognitive performances [1]. Mechanistically, previous studies on cerebral blood flow could show restrictions of cerebral blood flow in the resting period after exercise in the heat (−15% for a 1.5 °C increase in body core temperature [18]), although cerebral blood flow increased during exercise (about 20%, [18]) in general. 

However, comparable to studies on the impairment of exercise capacity, studies on cognitive performance in the context of dehydration use models of exercise-induced or heat-induced dehydration [1]. Therefore, it is difficult to differentiate between the effects of dehydration, changes in metabolic or electrolyte milieu, or post-exercise fatigue. 

Sports-related concussion is a relevant risk of contact sports with several million cases per year worldwide [19]. The term concussion originates from the Latin term “concussio”, which means to shake violently [20] and defines the mildest form of traumatic brain injury, typically without macroscopic signs of brain injury [21]. 

Concussions can occur in any sport when relevant force impacts an athlete’s body, especially to the head or neck [22], that causes impairment of cognitive function (e.g., behavioural changes or slowed reactions) or symptoms of neurological trauma like loss of consciousness, dizziness, or headache. Data about concussions in boxing or martial arts are sparse [20]. For sports with reported data, the highest risk sports are football, rugby, ice hockey, and wrestling for male athletes, and soccer and basketball for female athletes [20]. A definitive diagnosis of concussion is a complex process and cannot rely solely on a single test [22]. The Sport Concussion Assessment Tool (SCAT) is a widely used instrument that evaluates symptoms in combination with an assessment of cognitive functions like memory, coordination, and concentration [23]. Concussion is a clinical diagnosis, [24] and head imaging is not mandatory for the diagnosis but might have relevance for ruling out structural brain damage [22,24]. 

The prognosis of concussion is good, but repeated concussion, especially within a “window of susceptibility” after a previous concussion, may induce post-concussive syndrome or chronic neurodegeneration [21]. Therefore, immediate diagnosis of sports-related concussion is crucial to prevent sequelae. 

Nevertheless, it has been shown that concussion is underdiagnosed because of low awareness, as demonstrated in a recent survey by Bazo et al. on youth rugby players and their coaches [25]. A total of 1719 athletes and their coaches underwent a survey on knowledge about concussion. The median knowledge score was 55% in athletes and 60% in coaches. Only 33% of athletes and 40% of coaches were familiar with key facts like the higher risk of a second concussion after an undiagnosed or not adequately treated first concussion. 

The pathophysiology of concussion and its typical symptoms involves complex processes of neuro-inflammation and alterations of metabolism [26] and similar inflammatory processes are also described in the pathophysiology of extreme dehydration in the heat/heatstroke [27]. In the absence of a multiparametric assessment tool for dehydration, induced impairment of cognitive performance (like the SCAT questionnaire for concussion) and taking into consideration the possible similarity of clinical presentation, the diagnosis of concussion might be impeded by an overlap with dehydration-associated effects on cognitive performance. 

The aim of the current study was to elucidate the impact of isolated, mild dehydration (hypohydration) on concussion-like symptoms, memory, balance, and coordination in recreational athletes without a history of concussion undergoing short, high-intensity exercise testing. To establish a direct link between hypohydration and the observed effects, the study examined the effects in the absence of heat or exhaustive endurance exercise.

## 2. Methods

The current study is part of a prospective survey on the effects of dehydration on key aspects of sports-physiological like exercise capacity, cognitive performance, and cardiac burden. 

The results regarding exercise capacity, as well as a description of the study cohort and baseline characteristics, have been published previously [5].

Briefly, self-appointed, ambitious recreational athletes with an age range of 18–50 years were recruited from the medical staff of our hospital. The key inclusion criterion was a willingness to participate in our study; proof of prior exercise performance or participation in prior competitions was not required. Ambitious was defined as an average weekly exercise time of at least two hours during the previous 12 months but without a history of professional sports. No exclusion was made for specific sports. Athletes with any pre-existing medical conditions, especially traumatic brain injury or concussion in the past, were excluded. The study was approved by the local ethics committee (accession number: 22-0295) and conducted in accordance with the declaration of Helsinki. All participants provided written informed consent. 

We have previously described the model of hypohydration and the specific exercise test we used in our earlier work [5]. The athletes underwent the protocol twice (once in a euhydrated [EUH] and once in a hypohydrated [HYH] state). Hypohydration was achieved by undergoing a standardised fluid deprivation test for 12 h, as is used in endocrinological diagnostic workup of polydipsia [28,29]. The subjects were advised to refrain from drinking or eating meals with high fluid content (e.g., soup). No further provisions were made for food frequency, quantity, or composition. The euhydrated scenario was without any specific dietary regulation. Athletes neither underwent recording of diet nor standardized weighing as the expected loss of body mass was at a low level. 

The sequence of euhydrated and hypohydrated exercise was randomized using opaque sealed envelopes and took place at comparable times of day to rule out bias due to circadian rhythm. The interval between both exercise tests was between 7 and 14 days. Participants performed a cycle ergometer exercise test with a ramp protocol at ambient conditions in our department, starting with 40 watts (women) or 60 watts (men) with a target cadence between 60 and 70 rpm. Cardiopulmonary exercise testing (CPET) was conducted until exhaustion. The slope of the ramp protocol was calculated individually in accordance with the patient’s self-assessment targeting for an exercise time between 8 and 10 min. No difference in the CPET protocol was made between both study visits. 

Body composition, as a measure of dehydration, was estimated using the Nutribox body impedance analyser (Data Input, Germany), according to the manufacturer’s instructions. Nutribox measurements are based on bioelectrical impedance vector analysis with three relevant parameters [30]: resistance (R), as a measure of body water; reactance (Xc), as a measure of body cell mass, i.e., cell membranes; and finally, the corresponding phase angle (PhA), as a result of electrical phase shift of alternating current [30].

Sports concussion-like symptoms, memory, balance, and coordination were evaluated immediately after completion of the cardiopulmonary exercise test. We used the Sport Concussion Assessment Questionnaire (SCAT), 3rd version [23], omitting the Glasgow Coma Scale (SCAT item #1), play-specific aspects (SCAT item 2), and neck examination (SCAT item 5). 

Participants were asked about the following symptoms (SCAT item #3): headache, pressure in the head, neck pain, nausea or vomiting, dizziness, blurred vision, balance problems, sensitivity to light, sensitivity to noise, feeling slowed down, feeling like in a fog, not feeling right, difficulty concentrating, difficulty remembering, fatigue or low energy, confusion, drowsiness, trouble falling asleep, being more emotional, irritability, sadness, nervousness and anxiety. Besides the “total number of symptoms”, each present symptom was rated on a scale from 1 (mild) to 6 (severe) and summed to the “symptom severity score”. 

Cognitive assessment (SCAT item #4, 8) consists of questions with respect to:

Orientation (What month is it? What is the date today? What is the day of the week? What year is it? What time is it right now?), the resulting “Orientation Score” is between 0 and 5 correct answers. 

Immediate memory (three different lists of five items have to be repeated), the “immediate memory score total” is between 0 and 15 correct answers. Athletes were asked to repeat as many items as they can at the end of the SCAT evaluation (delayed recall). One point is given for each item mentioned, with a maximum score of 5. 

Concentration (four series of digits that have to be recalled backwards plus repetition of the month in the reverse order); the “Concentration Score” is between 0 and 5 correct series. 

Balance examination (SCAT item #6) contains Modified Balance Error Scoring System (BESS) testing [31]: athletes have to stand for 20 s with eyes closed and hands on their hips with both feet together (double leg stance), standing on the dominant foot (single leg stance), and with both feet heel-to-toe with the non-dominant foot in the back (tandem stance). Types of errors are: hands lifted off iliac crest; opening eyes; stepping, stumbling or falling; moving hip into >30 degrees abduction; lifting forefoot or heel; remaining out of test position >5 s. During the 20 s, all errors are summed up with a maximum “score” of 10. 

Additionally, tandem gait time [32,33] is measured as described in SCAT3 [23]: Participants are instructed to stand with their feet together behind a starting line (the test is best conducted with footwear removed). Then, they walk in a forward direction as quickly and as accurately as possible along a 38 mm wide (sports tape), 3 m long line with an alternate foot heel-to-toe gait, ensuring that they approximate their heel and toe on each step. Once they cross the end of the 3 m line, they turn 180 degrees and return to the starting point using the same gait. A total of 4 trials are carried out, and the best time is retained. Athletes should complete the test in 14 s. Athletes fail the test if they step off the line, have a separation between their heel and toe, or if they touch or grab the examiner or an object. In this case, the time is not recorded, and the trial is repeated, if appropriate [23]. The study workflow is depicted in Figure 1. 

## 3. Statistical Analysis

All data are presented as mean ± standard deviation of mean. Students’ *t*-test was used for comparison of continuous variables. *p*-values < 0.05 were considered statistically significant. All statistical analyses were performed using SPSS version 23 (IBM Corp., Armonk, NY, USA).

## 4. Results

Baseline characteristics and results of fluid deprivation assessment have been published in our previous work [5]. Briefly, participants were 17 women and 33 men. Mean age was 29.7 ± 6.9 years. A Body Mass Index of 20.2 ± 2.9 kg/m^2^ represents a cohort of ambitious recreational athletes. Mean height was 177 ± 10 cm, with a mean weight of 71.9 ± 12.5 kg (Table 1). Body water was decreased by 2% after fluid deprivation (39.44 ± 7.2 vs. 38.63 ± 7.0%; *p* < 0.01) with a concomitant decrease in PhA (5.83 ± 0.73 vs. 5.66 ± 0.71; *p* = 0.02), as described in our previous work. This corresponds to a calculated body mass loss of about 0.8%. Baseline characteristics as well as the results of the fluid deprivation test are depicted in Table 1.

Athletes reported significantly more symptoms in the hypohydrated state (1.8 ± 2.2 vs. 0.4 ± 0.7; *p* < 0.01). Symptoms reported were more severe with a significantly higher symptom severity score (4.4 ± 6.2 vs. 1 ± 1.9; *p* < 0.01, Figure 2).

Orientation was without relevant differences (HYH: 4.98 ± 0.1 vs. EUH: 5 ± 0; *p* = 0.3). Immediate memory testing revealed a discrete but significantly improved score in the hypohydrated state (14.8 ± 0.5 vs. 14.6 ± 0.7, *p* = 0.03). Concentration (HYH: 4 ± 0.8 vs. EUH: 4.1 ± 0.9; *p* = 0.7) and delayed recall score (HYH: 4.3 ± 0.8 vs. EUH: 4.3 ± 0.9, *p* = 0.8) were not altered (Figure 3). 

Testing of balance and coordination showed significantly more errors in the tandem stance in hypohydration (1.1 ± 1.3 vs. 0.6 ± 1.1; *p* = 0.02). With errors in the double leg stance (not shown, HYH: 0 ± 0 vs. EUH: 0 ± 0; *p* = n.a.), single leg stance (HYH: 1.7 ± 1.5 vs. EUH: 1.5 ± 1.2; *p* = 0.4), coordination score (not shown; HYH: 1 ± 0 vs. EUH: 1 ± 0; *p* = 1), and best time of tandem gait (HYH: 12 ± 2 vs. EUH: 11.9 ± 2.4; *p* = 0.6) without relevant differences (Figure 4).

## 5. Discussion

To the best of our knowledge, this is the first study evaluating the effects of isolated mild dehydration (hypohydration) on concussion-like symptoms, memory, balance, and coordination in healthy recreational athletes. The key finding is that mild dehydration (hypohydration; body mass loss < 1%) leads to measurable alterations of balance and concussion-like symptoms while memory and coordination are unaltered. 

The impact of dehydration on skills, sporting task performance, and cognitive performance has been studied previously in small cohorts with very inhomogeneous protocols and has retrieved conflicting findings [1]:

McGregor et al. studied the impact of dehydration on soccer skills and concentration in nine semi-professional soccer players. The participants completed a 90 min intermittent exercise protocol (Loughborough intermittent shuttle test [34]), either ingesting or abstaining from fluid intake. Soccer skill and concentration were assessed before and immediately after the exercise protocol. Soccer skills remained unchanged during the “fluid trial” but deteriorated significantly (by 5%) in the no-fluid setting [35].

Edwards et al. studied 11 moderately active soccer players. Three experimental conditions were tested: no fluid intake, fluid intake ad libitum, no fluid intake but mouth rinse. Subjects underwent a 45 min period cycle ergometer exercise test (at 90% of individual ventilatory threshold) before they had a 45 min soccer match. Afterwards subjects were to complete a soccer-specific fitness test (yo-yo intermittent recovery test [36]) and a test of concentration. They found significantly increased body core temperature accompanied by a significant reduction in soccer-specific fitness. Nevertheless, concentration was without differences between all experimental conditions [37]. 

Bowling velocity and accuracy were studied by Devlin et al. on seven experienced cricket players. The study protocol contained a euhydrated as well as a dehydrated scenario. The athletes had to complete a bowling test of 36 deliveries. Afterwards they had to complete one hour of exercise in a heated environment (with or without drinking) followed by a repetition of the bowling test. Body mass loss was 2.5 kg in the dehydrated and 0.5 kg in the euhydrated scenario. They found that force (represented by bowling velocity) was not altered whereas accuracy (represented by bowling line and length) deteriorated relevantly [38]. 

Szinnai et al. studied objective and subjective motor function in 16 participants. In a randomized cross-over design, testing was performed once after 24 h of fluid deprivation and once during regular fluid intake. Fluid deprivation caused a 2.6% loss of body mass. Neither cognitive motor function (paced auditory serial addition task, five-choice reaction time test, manual tracking test, word colour conflict test) nor neurophysiological function (auditory event-related potentials) differed relevantly. Interestingly, the subjective ratings of tiredness, reduced alertness, and perceived effort were increased in a dehydrated state. An interaction for sex was seen for concentration/reaction with improved test results in men and worsened results in women [39]. The extent of dehydration was less pronounced in our cohort; nevertheless, we could not confirm the influence of sex on the studied items of concentration in our larger study cohort. 

The finding of decreased ability to concentrate or reduced alertness was confirmed in a study by Shirreffs et al. on healthy individuals undergoing a 37 h fluid deprivation [40]. In addition, subjects reported more symptoms like headache, which is in line with our finding of more and more severe concussion-like symptoms in the SCAT3 questionnaire. 

Cian et al. examined the effects of variation in body hydration in eight healthy men. Besides a euhydrated control, subjects were dehydrated (by controlled passive hyperthermia or exercise up to a weight loss of 2.8%) or hyperhydrated by infusion of a solution containing glycerol. After a recovery period, subjects had to complete an ergometer exercise protocol before cognitive performance was tested. According to the previously mentioned studies, subjective estimates of fatigue were increased in both dehydrated conditions. Short-term memory was found to be worsened in subjects when dehydrated but improved significantly compared to euhydration when subjects were hyperhydrated [41]. In a following study with a comparable protocol of dehydration and assessment of cognitive performance, subjects received rehydration (solution with glucose and NaCl) for complete compensation of dehydration. The assessment of cognitive function was repeated two hours later. Cognitive impairment after dehydration and exercise was identified as a transient phenomenon as subjects did not show any alterations, neither rehydrated nor still dehydrated [42]. 

Sharma et al. studied the effects of heat-stress-induced dehydration on cognitive performance in eight heat-acclimatized inhabitants of tropical regions. The extent of dehydration studied was 1, 2, and 3% loss of body mass. The subjects had to exercise in hot dry (45 °C, 30% relative humidity) and hot humid (60% relative humidity) conditions. The very small study could not show significant alterations in routine mental work, although a trend was seen for 2 and 3% loss of body mass [43]. 

The effect of dehydration on concussion-like findings was studied by Collins et al. in a cohort of 16 recreational athletes, comparable to our cohort [19]. The protocol also used a shortened version of the SCAT3 questionnaire. Participants underwent evaluation in resting conditions (control) and after a 60 min bout of cycle ergometer training (65–70% age predicted calculated heart rate). The exercise protocol was performed once without fluid restriction (0 L/h) and once with fluid intake (1.0 L/h). No information is provided about the extent of dehydration in fluid-restricted scenarios or differences in exercise performance. The number of symptoms and the severity of symptoms were significantly higher when comparing the euhydrated session to the control (symptom total: 1.3 vs. 2.2; symptom severity: 1.6 vs. 3.2) and significantly higher under dehydration compared to euhydration (symptom total: 2.2 vs. 3.9; symptom severity: 3.2 vs. 6.2). The scores in our study in dehydration (symptom total: 1.8; symptom severity: 4.4) lay within the range between exercise in euhydration and exercise in dehydration in Collin’s study. Collin’s study shows that dehydration influences concussion-like symptoms independently from general fatigue after exercise. Our study could confirm this finding. 

In contrast to our study, Collin did not show significant effects on balance. With respect to the numerical trend (total BESS in control: 2.1; in euhydration; 2.9, in dehydration 4.5), this has to be interpreted as a lack of statistical power. The influence of dehydration on BESS was also documented in a study on collegiate wrestlers after weight-cutting [44]. The clinical relevance of these cognitive effects of dehydration is described in a study on the association of the in-competition injury risk and the degree of rapid weight- cutting in collegiate wrestlers [45]. In a cohort of 67 athletes, pre-competition weight loss was significantly higher in injured athletes compared to athletes without in-competition injury (−7% vs. 5.7%). The hazard for in-competition injury was 1.14 for every kilogram of body weight lost. 

The pathophysiological mechanism for impairment of cerebral performance remains elusive. One aspect might be dysregulation of cerebral blood flow (CBF). It has been shown that CBF rises about 20% under endurance exercise [18]. However, exercise in the heat with a concomitant rise in body core temperature might reduce CBF after exercise (15% for a 1.5 °C increase). Compensation for this is possible when fluid loss is replaced but might gain relevance in dehydration. The reduction in CBF during exercise in the heat has been shown especially in recreational athletes [18]. Transient (short-lasting) reduction in CBF after exercise in dehydrated athletes, as the critical factor for impairment of cognitive performance, might explain the findings of the above-mentioned studies by Cian et al. describing spontaneous regression within several hours even when dehydration was not compensated. Another aspect might be a reduction in cardiac output in the dehydrated state, as we demonstrated in our previous work [5]. It might be conceivable that this could aggravate the reduction in CBF even if it has to be taken into consideration that CBF is highly prioritized in circulation [18]. Finally, it has to be kept in mind that many of the instruments for measurement of cognitive performance are influenced by the subjective feelings of the athletes or their motivation. Therefore, it remains unclear whether these effects are attributable to dehydration in and of itself or (at least in part) to the negative psychological associations derived from a greater perception of effort in a dehydrated condition [37]. 

Impairment of cognitive performance in dehydration could be of relevance for different reasons. First, most sports and exercises involve some aspect of cognitive performance, mental readiness, reaction response, and motor control [1]; therefore, sports performance might suffer (e.g., Devlin et al. [38]). Second, impairment of balance, reaction, or alertness might increase the risk of injury during sports (Hammer et al. [45]). Third, in the context of concussion assessment, alterations of cognitive performance, as found in our study, could mimic concussion-like findings.

### Limitations and Perspectives

Our study included ambitious recreational athletes with various sports experiences. The shown effects of dehydration may or may not be transferable to professional athletes or persons with lower sports experience. It would be of interest to study the described effects in athletes’ sports specifically with respect to sports-specific test protocols. 

As expected, the loss of body mass was low, dehydration was assessed by BIA and not by other modalities like standard weighing. As a final point, the practical relevance of our findings for concussion assessment can only be answered somewhat as only a single assessment tool was used in our study, which was additionally used in third edition (with versions 4, 5, and 6 available) and without baseline testing before CPET. 

## 6. Conclusions

Mild dehydration caused relevant alterations of concussion-like items such as typical symptoms or balance in healthy recreational athletes in the absence of endurance exercise or heat. These findings are in contrast to prior studies yielding negative results in smaller cohorts. 

Further sports-specific research is needed to elucidate the real-life relevance of sports performance, injury risk as well as concussion assessment. 

## Figures and Tables

**Figure 1 nutrients-15-04420-f001:**
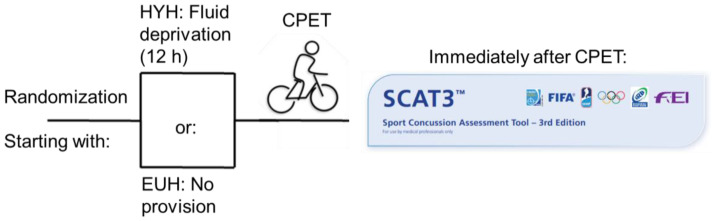
The figure depicts the study protocol: randomized start with either the EUH or HYH scenario. Participants were placed on a cycle ergometer with an individualized ramp protocol. SCAT3 was assessed immediately after completion of the exercise test.

**Figure 2 nutrients-15-04420-f002:**
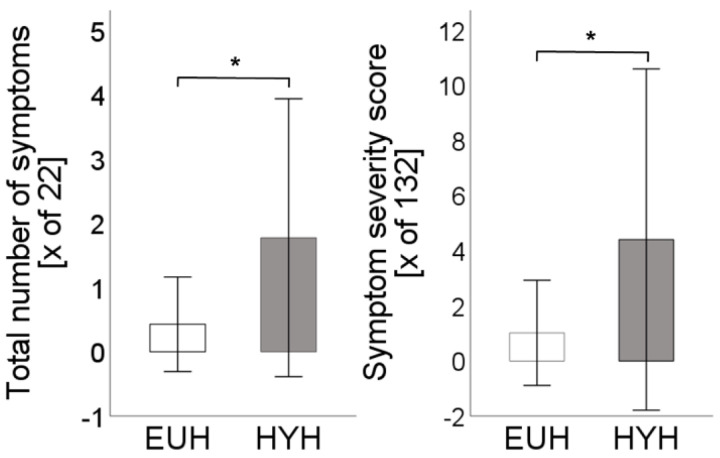
Total number of symptoms [x of 22] and symptom severity score [x of 132] are depicted for the euhydrated state [EUH, white bars] and hypohydrated state [HYH, grey bars], *: *p* < 0.01.

**Figure 3 nutrients-15-04420-f003:**
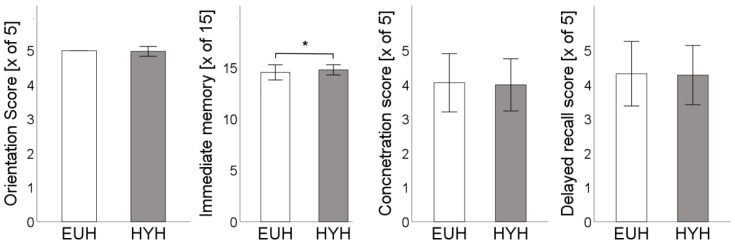
Orientation Score [x of 5], immediate memory score total [x of 15], concentration score [x of 5], and delayed recall score [x of 5] are depicted for the euhydrated state [EUH, white bars] and hypohydrated state [HYH, grey bars], *: *p* < 0.05.

**Figure 4 nutrients-15-04420-f004:**
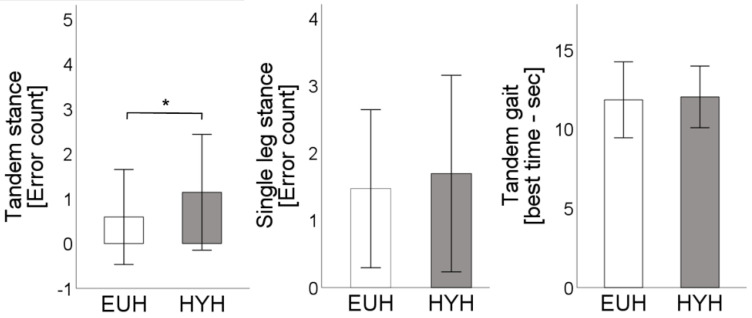
Error count in tandem stance and single leg stance as well as best time (out of four attempts) in tandem gait [sec.] are depicted for the euhydrated state [EUH, white bars] and hypohydrated state [HYH, grey bars], *: *p* < 0.05.

**Table 1 nutrients-15-04420-t001:** Baseline characteristics and results of the fluid deprivation assessment (data have been published previously [5]).

		n = 50
Age [years]		29.7 ± 6.9
Weight [kg]		71.9 ± 12.5
Height [cm]		177 ± 10
BMI [kg/m^2^]		20.2 ± 2.9
Female Sex [n]		17
Results of BIA	EUH	HYH
Bodywater [%]	39.44 ± 7.2	38.63 ± 7.0
Phase angle [°]	5.83 ± 0.73	5.66 ± 0.71
Calculated loss of body mass [%]	0.8	

## Data Availability

Data will be made available on reasonable request.

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
