# Peer review of "Impact of Preparticipation Hypohydration on Cognitive Performance and Concussion-like Symptoms in Recreational Athletes"

_nutrients, 2023, doi:10.3390/nu15204420_

Round 1

Reviewer 1 Report

The paper “Impact of Preparticipation Hypohydration on Cognitive Performance and Concussion-suspect Symptoms in Recreational Athletes” presents the relationship between mild dehydration caused relevant alterations and concussion-suspect symptoms and balance in healthy recreational athletes in absence of endurance exercise or heat. The study design is appropriate. The general results and conclusion described are consistent like most of literature on this theme.

Even if some major points should be addressed:

-          You used the review format, but I think it is better to modify it in original article.

-          Line numbers are missing.

-          Introduction: when you speak about citation 20, it would be better to substitute with the most recent one: Patricios JS, Schneider KJ, Dvorak J, et al Consensus statement on concussion in sport: the 6th International Conference on Concussion in Sport–Amsterdam, October 2022 British Journal of Sports Medicine 2023;57:695-711.

-          Introduction: concussion is underdiagnosed and with low awareness, it is important to underline the fact in relation with confounding factor as low hydratation pattern. As well described: Bazo, Marco; Arpone, Marta; Baioccato, Veronica MD; Ermolao, Andrea MD, PhD; Gregori, Dario PhD; Da Dalt, Liviana MD; Bressan, Silvia MD, PhD. Concussion Knowledge and Self-Reported Behaviors in Youth Rugby Players and Their Coaches: A Population-Wide Cross-Sectional Survey. Clinical Journal of Sport Medicine 33(5): p 541-551, September 2023. | DOI: 10.1097/JSM.0000000000001154

-          Methods: you should better justify the use of SCAT3, 10 years old multiparametric tool that owns many limits (as well described in current scientific literature). I think it would be added to limitations chapter.

-          Methods: as you know SCAT3 has interindividual variability, it would be useful to test a baseline (at rest) for each athlete. If you consider this assessment logistically impossible, better to add in limitations chapter or simply explain why.

-          Introduction and discussion: I think 2 concepts are missing, for a reader not usually involved in this research field.

o   The pathophysiological relationship between heatstroke, hypohydration and concussion related symptoms, related to brand new literature about the underlying inflammatory pattern.

o   The absence of a multiparametric assessment for heatstroke and hypohydration, and the reason why you choose SCAT3 to assess this condition.

-          A provocative question could be: “why do you need to differentiate hypohydration to concussion symptoms? I can easily relate concussion to trauma!”; maybe it would be useful to add a sentence related to this.

Minor editing of English language required

Author Response

Dear Reviewer,

Thank you very much for your comments on our manuscript. We revised our manuscript thoroughly according your suggestions. Please find enclosed our point-by-point reply to your comments:

Dear Reviewer,

Thank you very much for your comments on our manuscript. We revised our manuscript thoroughly according your suggestions. Please find enclosed our point-by-point reply to your comments:

The paper “Impact of Preparticipation Hypohydration on Cognitive Performance and Concussion-suspect Symptoms in Recreational Athletes” presents the relationship between mild dehydration caused relevant alterations and concussion-suspect symptoms and balance in healthy recreational athletes in absence of endurance exercise or heat. The study design is appropriate. The general results and conclusion described are consistent like most of literature on this theme.

Even if some major points should be addressed:

-          You used the review format, but I think it is better to modify it in original article.

-          Line numbers are missing.

The version of our manuscript you received for review underwent preliminary typesetting by the editorial office.  Our submitted manuscript contained line numbers and was formatted as “original research article”. We hope this issue will be easy to overcome within final typesetting.

-          Introduction: when you speak about citation 20, it would be better to substitute with the most recent one: Patricios JS, Schneider KJ, Dvorak J, et al Consensus statement on concussion in sport: the 6th International Conference on Concussion in Sport–Amsterdam, October 2022 British Journal of Sports Medicine 2023;57:695-711.

Thank you very much for this comment, we updated this citation.

-          Introduction: concussion is underdiagnosed and with low awareness, it is important to underline the fact in relation with confounding factor as low hydratation pattern. As well described: Bazo, Marco; Arpone, Marta; Baioccato, Veronica MD; Ermolao, Andrea MD, PhD; Gregori, Dario PhD; Da Dalt, Liviana MD; Bressan, Silvia MD, PhD. Concussion Knowledge and Self-Reported Behaviors in Youth Rugby Players and Their Coaches: A Population-Wide Cross-Sectional Survey. Clinical Journal of Sport Medicine 33(5): p 541-551, September 2023. | DOI: 10.1097/JSM.0000000000001154

This is an important aspect of the leading causes of underdiagnosis of concussion in sports.

We included this in the introduction section of our revised version of our manuscript.

“Nevertheless, it has been shown that concussion is underdiagnosed because of low awareness, as demonstrated in a recent survey of Bazo et al. on youth rugby players and their coaches (Bazo and others 2023). A total of 1719 athletes and their coaches underwent a survey on knowledge about concussion. The median knowledge score was 55% in athletes and 60% in coaches. Only 33% athletes and 40% of coaches were familiar with key facts like higher risk of a second concussion after a not diagnosed or not adequately treated first concussion. In addition, diagnosis of concussion might be impeded by an overlap with dehydration-associated effects on cognitive performance.”      

-          Methods: you should better justify the use of SCAT3, 10 years old multiparametric tool that owns many limits (as well described in current scientific literature). I think it would be added to limitations chapter.

Thank you very much for this important comment, we added this aspect to the limitation´s section.

The main reasons for us to use Version 3 of SCAT was good applicability and comparability to the study of Collins et al (as cited in the discussion section of our manuscript). Our hypothesis was that a lack of statistical power of their study was a relevant reason for most parameters staying unchanged.

We agree with you, that it might seem inappropriate to use version 3 of a test when version 6 is available. However, there has to be taken in consideration that, first, all items studied in our current trial are still components of SCAT6 and second, the main aspects of the evolution of SCAT (from version 3 to 6) like Assessment of “Concussion history” or ”immediate Assessment/Neuro Screen” are irrelevant for our study as presence of concussion was an exclusion criterion in our study. As described in the methods section of our manuscript items of SCAT questionnaire without relevance for our test setting were omitted. :

“Sports concussion-suspect symptoms, memory, balance, and coordination were evaluated immediately after completion of the cardiopulmonary exercise test. We used the Sport Concussion Assessment Questionnaire (SCAT) 3rd version [22], omitting the Glasgow Coma Scale (SCAT item #1), play-specific aspects (SCAT item 2), and neck examination (SCAT item 5).”

 In conclusion, we think the study would not have changed relevantly when using SCAT4, 5 or 6 questionnaire instead of SCAT3.

-          Methods: as you know SCAT3 has interindividual variability, it would be useful to test a baseline (at rest) for each athlete. If you consider this assessment logistically impossible, better to add in limitations chapter or simply explain why.

Thank you for raising this point. We agree with you that interindividual variability is an important issue in SCAT.

As we could demonstrate in our study, CPET without dehydration did not lead to pronounced alterations in SCAT (which was expected). Therefore, we decided to omit testing before CPET as we were afraid that four-time testing (twice in EUH and twice in HYH state) within 7-14 days could bias our results because of a “learning curve” of participants. Besides, each participant underwent the protocol twice and served as an own control.

Nevertheless, we added this limitation to our manuscript.

-          Introduction and discussion: I think 2 concepts are missing, for a reader not usually involved in this research field.

o   The pathophysiological relationship between heatstroke, hypohydration and concussion related symptoms, related to brand new literature about the underlying inflammatory pattern.

  • The absence of a multiparametric assessment for heatstroke and hypohydration, and the reason why you choose SCAT3 to assess this condition.

Thank you very much for this interesting two aspects. The cellular pathophysiology of cognitive performance in dehydration or concussion was not within the scope of our study. We added your two aspects of pathophysiology as well as the missing of an assessment tool for dehydration associated alterations to a new paragraph in the introduction section of the revised version or our manuscript.

“Pathophysiology of concussion and its typical symptoms involves complex processes of neuro inflammation and alterations of metabolism (Neumann and others 2023) and similar inflammatory processes are also described in pathophysiology of extreme dehydration in the heat/heatstroke) (Iba and others 2023). In absence of a multiparametric assessment tool for dehydration induced impairment of cognitive performance (like SCAT questionnaire for concussion) and taking into consideration the possible similarity of clinical presentation, diagnosis of concussion might be impeded by an overlap with dehydration-associated effects on cognitive performance.”

As you stated, no multiparametric assessment tool for dehydration does exist. Our study tried to elucidate effects of mild dehydration on concussion assessment.

-          A provocative question could be: “why do you need to differentiate hypohydration to concussion symptoms? I can easily relate concussion to trauma!”; maybe it would be useful to add a sentence related to this.

Trauma and dehydration are ever-present aspects of sports. Closing the diagnostic gap of concussion to reduce under diagnosis is important, nevertheless, unjustified disqualification of athletes will reduce acceptance of athletes and stakeholders (in professional sports). That is what makes the importance of valid discrimination between concussion and dehydration.

Reviewer 2 Report

The study is very interesting but several issues should be improved.

The idea of the paper is interestig, however it should be improved as the results are reflected in it.

First of all, it is more understandable for the reader to express the values of the variables in mean (standard deviation) than in mean (95% CI).

From my point of view, the introduction is well writen.

The methods section should be improved. Baseline characteristics should be included in this paper although you have previously published.  A table with basal data could be a good solution. The model of hypohydration should be also included. 

The data about body water after water deprivation should be considered as  a result of the research and should be included in the results section, not in the methods one. 

Figure 1 could be improved with less text and more graphical information.

The figures included in results sections are difficult to understand. 

Author Response

Dear Reviewer,

Thank you very much for your comments on our manuscript. We revised our manuscript thoroughly according your suggestions. Please find enclosed our point-by-point reply to your comments:

Round 2

Reviewer 1 Report

Dear Authors,

thanks for your response to my comments, I think your paper is now enough complete to be published. 

Kind regards

Author Response

Thank you so much

Reviewer 2 Report

Thank you for the changes made to the paper. There are still some small details that could be improved. First of all, in my opinion Table 1 could be better located in the results sections. Although these data have been already published, they are the baseline data of this study and part of the results.

The figures in results section are now more understables but I would eliminate the p value and only include an asterisk in the figure in case of significant differences (the value of p has already included in the text and there is not need to duplicate information).

Author Response

Dear Reviewer,

Thank you very much for your comments on our manuscript. We revised our manuscript thoroughly according your suggestions. Please find enclosed our point-by-point reply to your comments:

Thank you for the changes made to the paper. There are still some small details that could be improved. First of all, in my opinion Table 1 could be better located in the results sections. Although these data have been already published, they are the baseline data of this study and part of the results.

We included table 1 and the corresponding paragraph in the results section of our manuscript.

The figures in results section are now more understables but I would eliminate the p value and only include an asterisk in the figure in case of significant differences (the value of p has already included in the text and there is not need to duplicate information).

All figures were revised according to your suggestion. Thank you.